# Visual Blood, a 3D Animated Computer Model to Optimize the Interpretation of Blood Gas Analysis

**DOI:** 10.3390/bioengineering10030293

**Published:** 2023-02-25

**Authors:** Giovanna Schweiger, Amos Malorgio, David Henckert, Julia Braun, Patrick Meybohm, Sebastian Hottenrott, Corinna Froehlich, Kai Zacharowski, Florian J. Raimann, Florian Piekarski, Christoph B. Noethiger, Donat R. Spahn, David W. Tscholl, Tadzio R. Roche

**Affiliations:** 1Department of Anaesthesiology, University Hospital Zurich, University of Zurich, 8091 Zurich, Switzerland; 2Departments of Epidemiology and Biostatistics, Epidemiology, Biostatistics and Prevention Institute, University of Zurich, 8006 Zurich, Switzerland; 3Department of Anaesthesiology, Intensive Care, Emergency and Pain Medicine, University Hospital Wuerzburg, University of Wuerzburg, 97080 Wuerzburg, Germany; 4Department of Anaesthesiology, Intensive Care Medicine, and Pain Therapy, University Hospital Frankfurt, Goethe University Frankfurt, 60590 Frankfurt, Germany

**Keywords:** blood gas analysis, medical devices, point-of-care-testing, situational awareness, technology

## Abstract

Acid–base homeostasis is crucial for all physiological processes in the body and is evaluated using arterial blood gas (ABG) analysis. Screens or printouts of ABG results require the interpretation of many textual elements and numbers, which may delay intuitive comprehension. To optimise the presentation of the results for the specific strengths of human perception, we developed Visual Blood, an animated virtual model of ABG results. In this study, we compared its performance with a conventional result printout. Seventy physicians from three European university hospitals participated in a computer-based simulation study. Initially, after an educational video, we tested the participants’ ability to assign individual Visual Blood visualisations to their corresponding ABG parameters. As the primary outcome, we tested caregivers’ ability to correctly diagnose simulated clinical ABG scenarios with Visual Blood or conventional ABG printouts. For user feedback, participants rated their agreement with statements at the end of the study. Physicians correctly assigned 90% of the individual Visual Blood visualisations. Regarding the primary outcome, the participants made the correct diagnosis 86% of the time when using Visual Blood, compared to 68% when using the conventional ABG printout. A mixed logistic regression model showed an odds ratio for correct diagnosis of 3.4 (95%CI 2.00–5.79, *p* < 0.001) and an odds ratio for perceived diagnostic confidence of 1.88 (95%CI 1.67–2.11, *p* < 0.001) in favour of Visual Blood. A linear mixed model showed a coefficient for perceived workload of −3.2 (95%CI −3.77 to −2.64) in favour of Visual Blood. Fifty-one of seventy (73%) participants agreed or strongly agreed that Visual Blood was easy to use, and fifty-five of seventy (79%) agreed that it was fun to use. In conclusion, Visual Blood improved physicians’ ability to diagnose ABG results. It also increased perceived diagnostic confidence and reduced perceived workload. This study adds to the growing body of research showing that decision-support tools developed around human cognitive abilities can streamline caregivers’ decision-making and may improve patient care.

## 1. Introduction

Acid–base homeostasis is critical for all physiological processes [1]. Arterial blood gas (ABG) analysis is the diagnostic standard to detect imbalances in patients’ acid–base equilibrium, gas exchange, and electrolyte status [2,3,4]. The ability of modern ABG devices to measure a large variety of parameters from samples containing only a few millilitres of blood automatically, quickly, accurately, and repeatedly represents a tremendous technological achievement [5]. 

However, to take full advantage of the specific strengths of the human perceptual system, today’s ABG devices still have room for optimisation with respect to how they present diagnostic information. Today’s ABG result printouts consist of approximately 20 rows of tabular data, each including a parameter name, value, measurement unit, and expected normal range. Extracting information from such a printout requires caregivers to read and mentally translate a substantial quantity of textual and numerical data elements and integrate the derived meaning into their pre-existing mental models of ABG analysis. In these mental constructs, they assign the various parameters to specific physiological functions or abnormalities. These mental models vary widely among caregivers, which is reflected in the different orders in which caregivers read the parameters and in the different meanings they ascribe to them in different situations.

Moreover, this cognitively demanding process of interpretation happens in clinical environments where caregivers must deal with various factors that negatively affect their performance, such as information overload, distractions, and fatigue, which makes fostering situational awareness particularly challenging. Situational awareness is a three-step concept consisting of perceiving the relevant data elements in a situation, understanding their meaning, and projecting the situation’s significance into the near future [6,7]. High situational awareness allows us to make decisions that are optimally adapted to a given situation and to perform appropriate actions. Research has identified situational awareness breakdowns as the primary cause of adverse events in anaesthesia critical incident reporting system cases and malpractice claims [8,9], and has found strong evidence for a link between improving situational awareness and improving performance. For example, cognitive aids such as the World Health Organization safety checklists improve situational awareness and outcomes, and situational-awareness-oriented design improves diagnostic performance [10,11,12,13,14,15].

To optimise the information presented in the ABG results for achieving situational awareness, our research group developed Visual Blood (VB), an animated visualisation of the ABG results. This technology creates a virtual model of the ABG result, aiming to simplify the creation of caregivers’ own mental models. 

In this study, we reported the results of a multimodal quantitative and survey study of VB. We tested the primary hypothesis that participants make a higher number of correct diagnoses in simulated clinical scenarios with VB than with conventional ABG result printouts. We also examined the technology’s effects on participants’ perceived diagnostic confidence and workload. Furthermore, we evaluated the learnability of VB and conducted a survey to gain insights into caregivers’ acceptance of the concept.

## 2. Materials and Methods

The Cantonal Ethics Committee of Zurich, Switzerland issued a declaration of no objection (Business Management System for Ethics Committees Number Req-2021-00307). As this study was not subject to the Human Research Act, ethical approval was not required for the German centres. Study participation was voluntary and without compensation. We report this article following the CONSORT and STROBE guidelines for healthcare simulation research [16].

### 2.1. Visual Blood

Figure 1 and Appendix A explain VB in graphical and audio-visual format. We also provide a pixel stream of the program at shorturl.at/bHQRW. We developed VB using the gaming engine Unreal Engine 4 (Epic Games, Inc., Raleigh, NC, USA). The version evaluated in this study is a software prototype simulating ABG results and is yet to be interfaced with an ABG device.

VB is a computer animation showing a virtual model of any given ABG situation by visualising the ABG parameters and their interactions as intuitive 3D icons. We developed it based on the principles of user-centred and situational-awareness-oriented design [7] and our experience with previously developed visualisations [17,18,19,20,21,22,23,24,25]. It follows the goal of situational-awareness-oriented interface design: to convey the information needed by the caregivers as quickly as possible and with the lowest cognitive effort [7].

To be recognisable as a model of the ABG result and thus support global situational awareness, VB positions the caregivers’ viewpoint inside an arterial blood vessel. The individual parameter visualisations flow through this vessel. Like the artery, these visualisations have a logical commonality with the reality they intend to model. For example, in VB, high plasma osmolarity is indicated by a high number of H_2_O molecules flowing into the blood vessel through its wall. VB groups parameter visualisations according to their function or place of action in actual blood. For example, oxygen and carbon monoxide molecules bind to the erythrocytes’ oxygen-binding sites, as they would in reality. This aims to facilitate the simultaneous perception and understanding of the status of interconnected parameters. To make critical signals visually salient, we selected a logic for these animations that also has a meaningful relation to reality. Parameter visualisations that are too low become greyed-out, dashed and blinking, which are visual clues commonly used to indicate a missing or non-available part. Parameter visualisations that are too high appear in much higher numbers than when normal. The reduction of information complexity by classifying parameters as either too low, normal, or too high is a further situational-awareness-oriented design principle we used. In the conventional ABG result printouts, the care providers perform this classification themselves with the help of the normal reference values provided.

### 2.2. Study Design and Participants

This was an investigator-initiated, multicentre, randomised, prospective, computer-based, within-subject simulation study comparing VB and conventional ABG result printouts. Anaesthesiologists and intensivists in training (resident physicians), or already board-certified (staff physicians), were included. We recruited participants from three hospitals: the University Hospitals of Zurich in Switzerland and of Frankfurt and Wuerzburg in Germany. 

The study consisted of two parts. In the first part, we investigated how well the participants assigned individual VB visualisations to their corresponding ABG analysis parameters after a short educational video. In the second part of the study, we compared VB with conventional ABG result printouts, testing the hypothesis that using VB enables participants to improve their perception of individual parameters, make a higher number of correct diagnoses, perceive higher diagnostic confidence, and lower workloads. 

### 2.3. Study Procedure and Outcome Measures

We welcomed the participants, obtained written informed consent, and asked them to answer a brief demographic survey. Then, we presented the educational video “VB Education” (Appendix A). After watching the video, the participants had time to ask questions or rewatch video sections. 

For part one of the study, investigating the assignment of VB visualisations to their corresponding or intended ABG parameters (visualisation assignment), we showed participants 15 s long VB sequences, in each of which precisely one ABG parameter was outside the norm. We then asked which parameter it was, and in which direction (“too high” or “too low”). If a parameter and the direction of its deviation were correctly assigned, a correct assignment was counted for the visualisation in question. For 14 of the total 18 parameters, we displayed a VB sequence with values that were too low and too high. Parameters that deviated in a single direction of clinical relevance were displayed only in this direction (i.e., lactate, methaemoglobin, carboxyhaemoglobin, and oxygen saturation). Thus, there were 32 sequences per participant, which we presented in randomised order (ResearchRandomizer version 4.0; http://www.randomizer.org; accessed on 1 April 2022).

Part two of the study compared ABG results shown as VB representations to the same ABG results shown as conventional ABG result printouts, evaluating the primary outcome—correct clinical diagnosis—as a binary variable.

We showed the participants scenarios with multiple deviations, each matching a clinical diagnosis. The scenarios always lasted 15 s, after which the screen turned black. We showed 2 × 3 scenarios to each participant, once with VB and once as a conventional ABG result printout, enabling three within-subject comparisons per participant. After each scenario, we requested the participants assign the displayed ABG results to 1 of 12 clinical diagnoses and to rate for all ABG parameters whether they were too low, within normal range, or too high. In addition, participants rated, after each scenario, their diagnostic confidence and perceived workload. The three presented scenarios originated from a pool of six different scenarios. Scenario selection, order, and type of presentation (VB or conventional ABG result printout) were randomised. As a secondary outcome, we examined the difference in clinical diagnosis performance between VB and conventional ABG printout as a function of participants’ individual performance with conventional ABG. In doing so, we aimed to evaluate whether VB can particularly support caregivers who perform less well in conventional ABG.

A further secondary outcome was parameter perception, defined as the number of correctly perceived ABG parameters. Participants had to indicate the status of each parameter visualisation (18 in each scenario). Further secondary outcomes included diagnostic confidence and perceived workload. The participants rated their subjective diagnostic confidence on a 4-point Likert scale (from very unconfident to very confident) and perceived workload using the National Aeronautics and Space Administration Task Load Index (NASA-TLX; from 0–100) [26,27,28]. 

Before leaving, participants rated four general statements on a 5-point Likert scale (from strongly disagree to strongly agree) to capture their impressions about VB. All responses were entered into a survey (Harvest your data, Wellington, New Zealand) on an iPad (Apple Inc., Cupertino, CA, USA) [29].

### 2.4. Statistical Analysis 

For descriptive statistics, we show medians and interquartile ranges for continuous data and numbers and percentages for categorical data.

To analyse the outcome of part one of the study—visualisation assignment—we used mixed logistic regression models with a random intercept per participant and the respective parameter as covariate to estimate the proportion of correct visualisation assignments (correct assignment of parameters and parameter deviations), considering that we had repeated measurements of the same persons that were not independent. We show the results as estimated percentages with confidence intervals.

To analyse the outcomes of part two of the study—correct clinical diagnosis, parameter perception and diagnostic confidence—we also used mixed logistic regression models with a random intercept per participant. To further explore the difference in clinical diagnosis performance between VB and conventional ABG printout, we analysed the participants’ individual differences between the number of correct answers using conventional ABG or VP in the same scenario using a linear mixed model with random intercept per participant. Note that in this case we ignored the ordering of the two methods (i.e., whether VP or ABG came first), but this should not have much effect because of the randomization. To analyse the overall NASA-TLX, which represents perceived workload as a continuous outcome, we also ran a linear mixed model with random intercept for each participant. All models were adjusted for potentially relevant covariates, such as age, gender, work experience, study centre, and scenario [16]. 

### 2.5. Sample Size Calculation

The sample size was calculated using data from part one of the study, i.e., the estimated proportion of correctly assigned visualisation parameters. We calculated that in order to construct a 95% confidence interval for an estimated proportion that extends no more than 10% in either direction, 35 participants were needed if the true proportion was 90%, and 62 participants were needed if the true proportion was 80%, so that our final sample size of 70 was enough to obtain the desired precision. 

To generate the statistical report, we used R version 4.0.5 (R Foundation for Statistical Computing, Vienna, Austria). A *p*-value <0.05 was considered to indicate statistical significance.

## 3. Results

Between April and May 2022, we recruited a total of 70 participants drawn from the three study centres’ anaesthesia staff. Table 1 displays participant characteristics.

### 3.1. Study, Part One

In the first part of the study, we investigated visualisation assignment and found that across all participants and all tested parameter visualisations and deviations of VB, 2011/2240 (90%) visualisations were correctly assigned. The mixed logistic regression model yielded an estimated proportion of correct visualisation assignments per participant of 91.5% (95%CI, 89.5% to 93.2%). To illustrate which parameter visualisations were particularly well or less well assigned, we show the number and percentages of correct visualisation assignment per parameter and the results of the mixed logistic regression model per parameter in Table 2.

### 3.2. Study, Part Two

In the second part of the study, we compared VB with conventional ABG result printouts. Figure 2 shows the study results on an individual participant level. Figure 3 shows the regression model results. 

#### 3.2.1. Clinical Diagnoses

Regarding our primary outcome, the correct diagnosis was made 180/210 (86%) times using VB compared to 142/210 (68%) times using the conventional ABG result printouts. The mixed logistic regression model showed an odds ratio (OR) of 3.4 (95%CI 2.00 to 5.79, *p* < 0.001) in favour of VB (Figure 2a).

#### 3.2.2. Parameter Perception

When participants used VB, more parameters were correctly perceived (3293/3780, 87%) versus conventional ABG result printouts (2915/3780, 77%). The mixed logistic regression model showed more than twice the odds (OR 2.16, 95%CI 1.90 to 2.46, *p* < 0.001) that participants perceived parameters correctly when they used VB instead of conventional ABG result printouts (Figure 2a).

The mixed linear model showed that participants with lower performance when using conventional ABG printouts benefited most from VB. The difference in the number of correctly recognized parameters using conventional ABG results versus VB increased by one point for every point that the conventional ABG diagnostic performance was lower (coefficient 0.96, 95%CI −0.80 to −1.04).

#### 3.2.3. Diagnostic Confidence

The mixed logistic regression model for assessing perceived diagnostic confidence (Figure 2a) showed that it was higher when participants used VB versus conventional ABG result printouts (OR 1.88, 95%CI 1.67 to 2.11, *p* < 0.001). 

#### 3.2.4. Perceived Workload 

The linear mixed model for assessing perceived workload (Figure 2b) showed that it was lower when participants used VB versus conventional ABG result printouts (coefficient −3.2, 95%CI −3.77 to −2.64). 

#### 3.2.5. Participant Opinions about VB

Fifty-one of seventy (73%) participants agreed or strongly agreed that VB was easy to use, fifty-five of seventy (79%) agreed that it was fun to use, and forty-two of seventy (60%) agreed that they would use it as an add-on to a conventional presentation of ABG results. Fifty of seventy (71%) participants agreed or strongly agreed that they thought VB would become part of their clinical routine. Figure 4 illustrates the four statements sampling participants’ general opinions about VB. 

## 4. Discussion

The conducted multimodal study investigated VB, a technology visualising ABG results which our research group developed to improve caregivers’ situational awareness in ABG interpretation. The study found that VB enabled participants to correctly diagnose more ABG cases, with a higher perceived diagnostic confidence and a reduced perceived workload compared to a conventional ABG result printout. Furthermore, the participants could easily learn to recognise the visualisations from a short educational video and rated their experience positively.

In two visualisation technologies we previously developed, Visual Patient for patient monitoring and Visual Clot for rotational thromboelastometry, we found the positive effects observed in computer-based simulation studies [17,20,21,22,23,24,25,30] to persist in high-fidelity simulation studies [18,31,32,33]. The substantial effect sizes observed to date with these technologies demonstrates their significant potential to improve decision-making. 

In this study, we also found that VB improved perceived diagnostic confidence, was easy to learn, and was most helpful to participants who had fewer correct diagnoses with the conventional ABG result printouts and who were younger. These results suggest that VB has the potential to shorten new caregivers’ education times and enable all members of a care team to participate in clinical decision-making. This aspect seems particularly important as projections from the World Health Organization show ever-increasing patient numbers, case complexity, and a global shortage of healthcare professionals [34,35,36].

The underlying mechanism for the improvements found with VB was likely its situational-awareness-oriented design, which focused on facilitating the formation of caregivers’ mental models by giving them virtual models of the ABG results. For example, in VB, a high number of lumps of sugar flowing through the artery indicates a high blood glucose level. This representation reduces caregivers’ cognitive effort, as the translation and classification of text abbreviations and associated numbers and their integration into caregivers’ own mental model are no longer necessary. Streamlining this mental model creation process was likely also responsible for the improved confidence and reduced perceived workload observed with VB. Pooled analyses found that task performance was proportional to perceived diagnostic confidence and inversely proportional to workload [28,37]. Justified diagnostic confidence is vital because clinicians who feel justifiably confident can make better and faster decisions. They do not need to second-guess and can react quickly and decisively to implement the consequences of correctly measured blood parameters [38].

In everyday life, information technology has advanced from email- and text-based to the image- and video-based internet. The quote “a picture is worth a thousand words (or numbers)” is also true in the medical domain [9]. As scientists and developers, we should aim to make clinicians’ work easier and enable them to develop their full potential. The positive feedback from the study participants who indicated that they would use VB in the future shows that they also sensed its potential. 

We designed VB to leave the final diagnosis to the human decision-makers. We provided the caregivers with visualisations to efficiently present the information but not to make a definitive diagnosis for them, as would be the case with a text output of the diagnosis.

Ultimately, combining both modalities—conventional ABG result printout and VB— will allow caregivers to make the best decisions. VB quickly and easily draws the caregivers’ attention to and promotes understanding of complex pathophysiologic relationships between parameters. Reading a more accurate number can provide more precise numerical information when needed.

A recent study has shown that visualisations that have a logical relation to reality can be recognised even without an instructional video [31]. This quality is desirable because there is often little time to obtain explanations in real life.

This study has strengths and limitations. Computer-based studies can indicate a technology’s functionality and potential benefits. They are particularly suitable for investigating the pure contribution of a technology to interpretation in isolation from distracting influences present in more complex situations. Ultimately, because reality is complex, the results of computer-based studies should be validated by high-fidelity simulation and real-life studies.

To make the scenarios unambiguous for the study participants, the degree of deviation they contained was more significant than that usually found in clinical practice. The gender- and experience-balanced selection of participants in three study sites spanning two countries minimised selection bias. Some visualisations in VB were more recognisable than others, which is why they need further refinement. An increase in recognisability may improve the performance of future VB versions. 

Furthermore, VB provides an impression of the biological imbalance without quantifying its intensity. However, in clinical practice, as described above, there are situations in which caregivers also need to refer to the numbers to determine the extent of the biological imbalance to treat the patient correctly. For this reason, we propose using VB and a numerical presentation of ABG results side by side.

## 5. Conclusions

In this study, VB enabled caregivers to correctly interpret more ABG results with little prior training (OR 3.4, 95%CI 2.00–5.79, *p* < 0.001) and increased their perceived diagnostic confidence (OR 1.88, 95%CI 1.67–2.11, *p* < 0.001). The study adds to the growing body of research showing that decision-support tools developed around our human abilities can streamline caregivers’ decision-making and help providers reach their full potential.

## Figures and Tables

**Figure 1 bioengineering-10-00293-f001:**
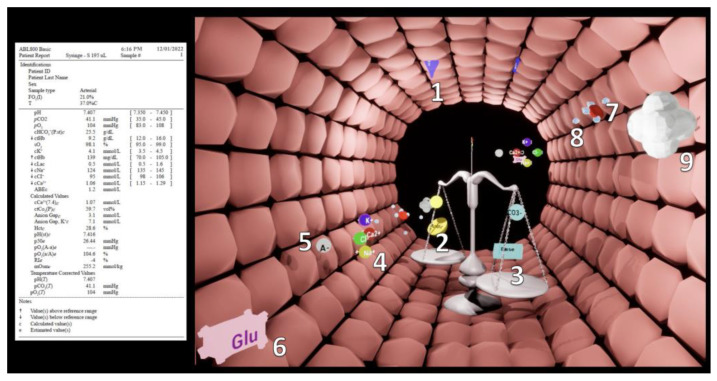
A standard arterial blood gas result printout on the left side and a VB visualisation on the right side. The blue water droplets diffusing through the vessel wall represent plasma osmolarity (**1**). The scale represents the acid–base balance. Acidic components include lactate and protons (**2**); alkaline components include bicarbonate and base excess (**3**). The electrolytes (**4**) sodium, potassium, chloride and calcium are illustrated in colour-coded form. The anion gap (**5**) is shown as a grey anion. Glucose (**6**) is illustrated in the form of pink sugar cubes. Haemoglobin (**7**) is represented by a red blood cell. It visualises together with oxygen molecules, oxygen saturation, and oxygen affinity, and can indicate the presence of too-high methaemoglobin or carboxyhaemoglobin concentrations. The unbound oxygen molecules represent the oxygen partial pressure (**8**). Carbon dioxide partial pressure (**9**) is shown in the form of a grey cloud.

**Figure 2 bioengineering-10-00293-f002:**
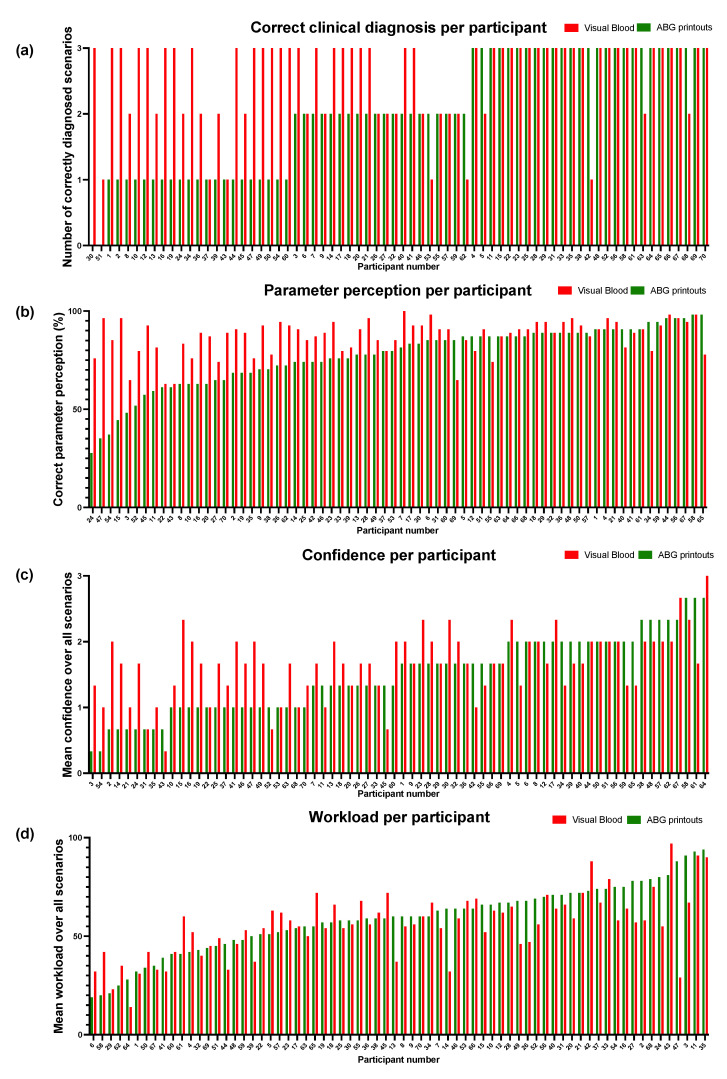
The study results on an individual participant level for correct clinical diagnosis, parameter perception, perceived confidence, and workload. The 70 participants are ranked on the *x*-axis from left to right in ascending order according to their achieved diagnostic and parameter perception performance, perceived confidence, or workload, respectively, with conventional ABG printout.

**Figure 3 bioengineering-10-00293-f003:**
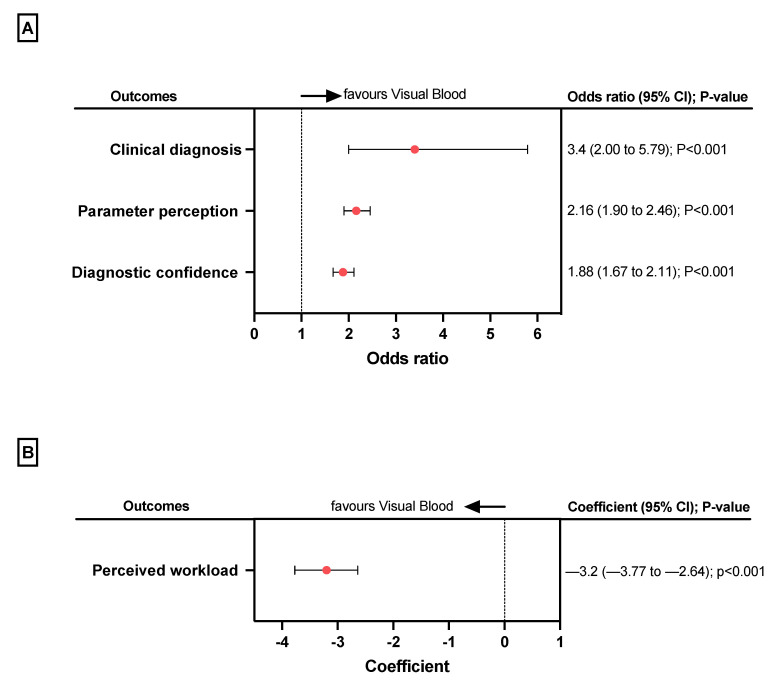
The results of the second part of the study. (**A**) The forest plots present the results of the mixed logistic regression models with a random intercept per participant for the outcomes of clinical diagnoses, parameter perception, and diagnostic confidence (**B**). The forest plot presents the linear mixed model results for the outcome of perceived workload. A negative coefficient means a lower workload using VB. CI, confidence interval.

**Figure 4 bioengineering-10-00293-f004:**
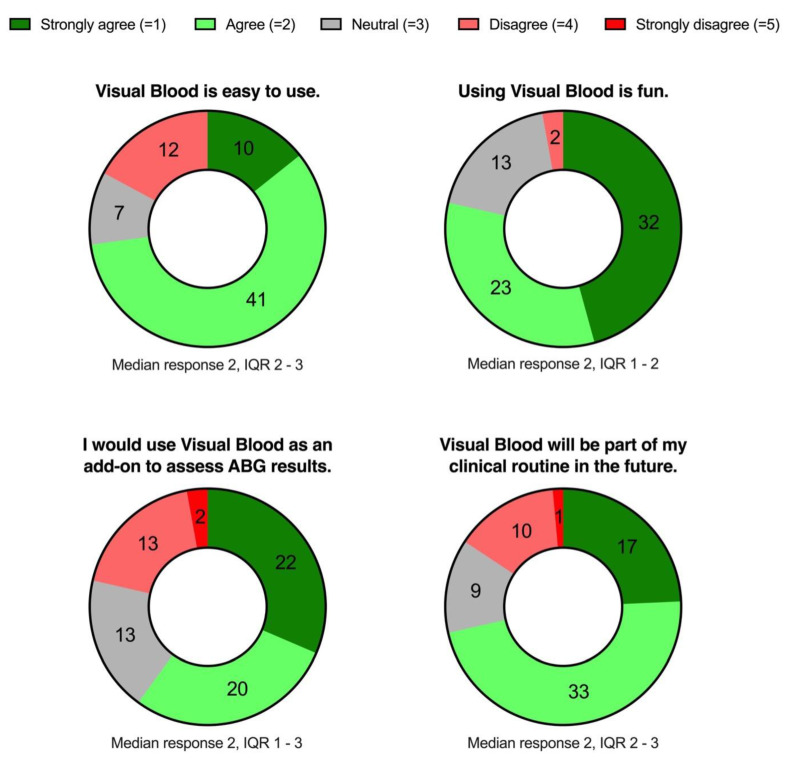
Doughnut charts presenting the participant opinions concerning Visual Blood. IQR, interquartile range. N = 70.

**Table 1 bioengineering-10-00293-t001:** Participant characteristics. USZ, University Hospital Zurich; UKW, University Hospital Wuerzburg; UKF, University Hospital Frankfurt; ABG, arterial blood gas; IQR, interquartile range.

Participants, n/N (%)	70
Participants from USZ, n/N (%)	35 (50)
Participants from UKW, n/N (%)	18 (26)
Participants from UKF, n/N (%)	17 (24)
Gender, female, n (%)	42 (60)
Resident physicians, n (%)	55 (79)
Staff physicians, n (%)	15 (21)
Participant age in years, median (IQR)	31 (28 to 35)
Work experience in years, median (IQR)	3.5 (2 to 6)
Self-rated theoretical ABG skills (0 = novice, 100 = expert), Median (IQR)	70 (58 to 78)

**Table 2 bioengineering-10-00293-t002:** Correct visualisation assignments per parameter (=parameter and deviation correctly assigned). Column two shows the numbers and percentages of correct visualisation assignments. Column three shows the estimated proportions according to the mixed logistic regression model per parameter, which considered repeated non-independent measurements from the same persons.

Parameter	Number (Percent) of Correct Visualisation Assignment	Estimated Proportion (95%CI) of Correct Visualisation Assignment
Lactate	70 of 70 (100)	1.00
Methaemoglobin	70 of 0 (100)	1.00
Partial pressure of carbon dioxide	140 of 140 (100)	1.00
Osmolarity	127 of 140 (91)	0.99 (0.96 to 1.00)
Glucose	138 of 140 (99)	0.99 (0.97 to 1.00)
Anion gap	133 of 140 (95)	0.98 (0.95 to 0.99)
Chloride	137 of 140 (98)	0.98 (0.95 to 0.99)
Potassium	137 of 140 (98)	0.98 (0.95 to 0.99)
Calcium	134 of 140 (96)	0.97 (0.94 to 0.99)
Bicarbonate	134 of 140 (96)	0.97 (0.93 to 0.99)
Sodium	133 of 140 (95)	0.96 (0.92 to 0.98)
Carboxyhaemoglobin	64 of 70 (91)	0.94 (0.85 to 0.97)
pH value	115 of 140 (82)	0.94 (0.89 to 0.97)
Haemoglobin	121 of 140 (86)	0.89 (0.82 to 0.93)
Partial pressure of oxygen	109 of 140 (78)	0.87 (0.80 to 0.92)
P50	105 of 140 (75)	0.86 (0.78 to 0.91)
Base excess	104 of 140 (74)	0.78 (0.69 to 0.85)
Oxygen saturation	40 of 70 (57)	0.58 (0.44 to 0.70)

## Data Availability

The data presented in this study are available on request from the corresponding author.

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
