# Peer review of "Visual Blood, a 3D Animated Computer Model to Optimize the Interpretation of Blood Gas Analysis"

_bioengineering, 2023, doi:10.3390/bioengineering10030293_

Round 1
Reviewer 1 Report
Thank you for sharing, for review the manuscript “Visual Blood, a 3D animated computer model to optimize the interpretation of blood gas analysis”.
The manuscript is adequately written, contains no ambiguity, and is not confusing. Its strengths are mainly its innovative aspect and focus on improving situational awareness to reduce perceptual-related accidents.
Here are some suggestions
Figure 1 is difficult to interpret. At first glance, it gives the impression of a progression over time, which is not the case. The difference between VB and conventional ABG should be highlighted differently (scatter plot?)
Figure 4 does not seem necessary
One of the limitations (to be discussed in the “limitations” section) is that the VB gives an idea of the biological imbalance without quantifying its intensity. In this simulation, nonclinical study VB could reduce cognitive load. However, in clinical practice, it could add a cognitive overload to interpret a clinical situation. Indeed, after looking at the VB, the clinician should still consult the numbers to determine the extent of the biological imbalance. The VB could be interesting in clinical practice when its results are balanced (without abnormalities). The clinician could eliminate the imbalance without deploying a high cognitive load in this case.
I hope this can help.
Issam.
Author Response
Review Report (Reviewer 1)
Comments and Suggestions for Authors
Thank you for sharing, for review the manuscript “Visual Blood, a 3D animated computer model to optimize the interpretation of blood gas analysis”.
The manuscript is adequately written, contains no ambiguity, and is not confusing. Its strengths are mainly its innovative aspect and focus on improving situational awareness to reduce perceptual-related accidents.
Here are some suggestions
Response
Dear reviewer,
Thank you very much for taking the time to study our manuscript in such detail and for giving us such good suggestions and criticism that we can improve the quality of the manuscript tremendously.
In the following, we have addressed each comment individually and have tried to incorporate them into the manuscript.
We hope that we have been able to respond to all of your comments to your satisfaction. If there are still any unclear points or further suggestions for improvement, please let us know.
We thank you for your excellent work on our manuscript.
Comment 1
Figure 1 is difficult to interpret. At first glance, it gives the impression of a progression over time, which is not the case. The difference between VB and conventional ABG should be highlighted differently (scatter plot?)
Response
I think you are referring to figure 2; figure 1 is the figure with the ABG printout and the image of Visual Blood.
In figure 2, we chose this type of result display to show the study results on an individual participant level. We have performed an interparticipant comparison, meaning that one and the same participant solved the same scenario, once with Visual Blood and once with an ABG printout. The diagram in Figure 2 makes it possible to see on an individual participant level which participant performed particularly well with which presentation technology. For example, you can see that participants who have difficulty recognizing the correct diagnosis with ABG printout do much better with Visual Blood. A scatter plot is, of course, also a popular way of result presentation, but in our case, it would not allow the comparison on an individual level. Even if it is a bit more complicated, the interested reader can get a lot of information from Figure 2, and we would like to keep it in the manuscript.
Nevertheless, we learned from your valuable comment that this figure could mislead the reader, and therefore we added “in ascending order” in the figure description for clarification.
Comment 2
Figure 4 does not seem necessary
Response
It is very important for us to show the subjective opinion of the participants about the technology, as this is essential for the willingness to use the technology in the future.
In our opinion, the doughnut charts seem to be a good way to present the results in a clear, easy-to-understand way, avoiding long text explanations and supplementing the study's main results.
Comment 3
One of the limitations (to be discussed in the “limitations” section) is that the VB gives an idea of the biological imbalance without quantifying its intensity. In this simulation, nonclinical study VB could reduce cognitive load. However, in clinical practice, it could add a cognitive overload to interpret a clinical situation. Indeed, after looking at the VB, the clinician should still consult the numbers to determine the extent of the biological imbalance. The VB could be interesting in clinical practice when its results are balanced (without abnormalities). The clinician could eliminate the imbalance without deploying a high cognitive load in this case.
Response
This is a perfectly valid point. We have implemented your suggestion in the discussion in the strengths and limitations section (p. 11 l. 365ff).
Reviewer 2 Report
This manuscript entitled “Visual Blood, a 3D animated computer model to optimize the interpretation of blood gas analysis.” developed Visual Blood to optimize the presentation of the results for the specific strengths of human perception. The authors bring an interesting study, but there are still some problems that cannot up this article to a publishing level. Suggestions are listed in the specific comments below.
Specific comments:
1. In the abstract part, please simplify the description of background of the article.
2. In the abstract part, please also add the relevant description about the contributions for future clinical or scientific research.
3. In the introduction part, line 43-46, “The ability of modern ABG devices to measure a large variety of parameters from samples containing only a few millilitres of blood automatically, quickly, accurately and repeatedly represents a tremendous techno-logical achievement.” Please cite relevant papers.
4. In the introduction part, line 76, “In this study, we report the results of a multimodal quantitative…” please write it in the past tense.
5. In the Materials and Methods part, Sample size calculation, line 221, “To create figures and graphs, we used GraphPad Version…”it is unnecessary to provide information about the software you used to generate figures.
6. In the discussion part, line 311-312. “The substantial effect sizes observed to date with these technologies demonstrates their significant potential to improve decision-making.” In the opinion of the reviewer, this sentence should not be a separate paragraph.
7. In the discussion part, 311-312, please also cite relevant references here. Some recently studies could also be added in the discussion, such as:
Current Status of Measurement Accuracy for Total Hemoglobin Concentration in the Clinical Context. Biosensors 2022, 12, 1147. https://doi.org/10.3390/bios12121147
Maternal Physical Activity and Neonatal Cord Blood pH: Findings from the Born in Bradford Pregnancy Cohort. Physical Activity and Health, 4(1), 150–157. DOI: http://doi.org/10.5334/paah.66
8. In the conclusion part, please also show detailed findings about this manuscript.
Author Response
Review Report (Reviewer 2)
Comments and Suggestions for Authors
This manuscript entitled “Visual Blood, a 3D animated computer model to optimize the interpretation of blood gas analysis.” developed Visual Blood to optimize the presentation of the results for the specific strengths of human perception. The authors bring an interesting study, but there are still some problems that cannot up this article to a publishing level. Suggestions are listed in the specific comments below.
Specific comments:
Response
Dear Reviewer, thank you for taking the time to read our manuscript and for helping to improve it significantly with your comments. We hope that we have been able to respond to all of your comments to your satisfaction and believe that readers will now receive an excellent paper.
Comment 1
- In the abstract part, please simplify the description of background of the article.
Response
Thank you for this comment. We have adapted the background part of the abstract (p.1 l.17ff.) to simplify its understanding.
Comment 2
- In the abstract part, please also add the relevant description about the contributions for future clinical or scientific research.
Response
We have rewritten the conclusion part of the abstract and have integrated your excellent suggestions (p. 1, l. 34ff.).
Comment 3
- In the introduction part, line 43-46, “The ability of modern ABG devices to measure a large variety of parameters from samples containing only a few millilitres of blood automatically, quickly, accurately and repeatedly represents a tremendous techno-logical achievement.” Please cite relevant papers.
Response
Thank you for reading the manuscript so carefully. The selected reference must have been lost here. We have now referenced this part of the introduction accordingly (p. 2, l. 48).
Comment 4
- In the introduction part, line 76, “In this study, we report the results of a multimodal quantitative…” please write it in the past tense.
Response
Thank you very much for pointing out this mistake. We have adjusted the sentence to the correct tense (p. 2, l.78).
Comment 5
- In the Materials and Methods part, Sample size calculation, line 221, “To create figures and graphs, we used GraphPad Version…”it is unnecessary to provide information about the software you used to generate figures.
Response
Thank you very much for this advice. We have omitted this information.
Comment 6
- In the discussion part, line 311-312. “The substantial effect sizes observed to date with these technologies demonstrates their significant potential to improve decision-making.” In the opinion of the reviewer, this sentence should not be a separate paragraph.
Response
Thank you very much for this formatting point. We have adjusted the paragraph as you suggested (p. 10, l. 312).
Comment 7
- In the discussion part, 311-312, please also cite relevant references here. Some recently studies could also be added in the discussion, such as:
Current Status of Measurement Accuracy for Total Hemoglobin Concentration in the Clinical Context. Biosensors 2022, 12, 1147. https://doi.org/10.3390/bios12121147
Maternal Physical Activity and Neonatal Cord Blood pH: Findings from the Born in Bradford Pregnancy Cohort. Physical Activity and Health, 4(1), 150–157. DOI: http://doi.org/10.5334/paah.66
Response
Thank you for providing the interesting studies. We have integrated the study by Stavchenko et al. into our manuscript (p. 10, l. 332-334).
Comment 8
- In the conclusion part, please also show detailed findings about this manuscript.
Response
In the conclusion, we now report the corresponding odds ratios and confidence intervals for our main findings (p. 11, l. 370-375).
Round 2
Reviewer 2 Report
All my questions have been well addressed, I recommend to accept now.